# Profile of Membrane Cargo Trafficking Proteins and Transporters Expressed under N Source Derepressing Conditions in *Aspergillus nidulans*

**DOI:** 10.3390/jof7070560

**Published:** 2021-07-14

**Authors:** Sofia Dimou, Xenia Georgiou, Eleana Sarantidi, George Diallinas, Athanasios K. Anagnostopoulos

**Affiliations:** 1Department of Biology, National and Kapodistrian University of Athens, Panepistimioupolis, 15784 Athens, Greece; sodimou@biol.uoa.gr (S.D.); geoxenia99@gmail.com (X.G.); 2Division of Biotechnology, Biomedical Research Foundation of the Academy of Athens (BRFAA), 11527 Athens, Greece; eleanasarantidi@gmail.com; 3Institute of Molecular Biology and Biotechnology, Foundation for Research and Technology, 70013 Heraklion, Greece

**Keywords:** *A. nidulans*, fungi, proteome, nutrient transporters, secretion, turnover

## Abstract

Solute and ion transporters are proteins essential for cell nutrition, detoxification, signaling, homeostasis and drug resistance. Being polytopic transmembrane proteins, they are co-translationally inserted and folded into the endoplasmic reticulum (ER) of eukaryotic cells and subsequently sorted to their final membrane destination via vesicular secretion. During their trafficking and in response to physiological/stress signals or prolonged activity, transporters undergo multiple quality control processes and regulated turnover. Consequently, transporters interact dynamically and transiently with multiple proteins. To further dissect the trafficking and turnover mechanisms underlying transporter subcellular biology, we herein describe a novel mass spectrometry-based proteomic protocol adapted to conditions allowing for maximal identification of proteins related to N source uptake in *A. nidulans*. Our analysis led to identification of 5690 proteins, which to our knowledge constitutes the largest protein dataset identified by omics-based approaches in *Aspergilli*. Importantly, we detected possibly all major proteins involved in basic cellular functions, giving particular emphasis to factors essential for membrane cargo trafficking and turnover. Our protocol is easily reproducible and highly efficient for unearthing the full *A. nidulans* proteome. The protein list delivered herein will form the basis for downstream systematic approaches and identification of protein–protein interactions in living fungal cells.

## 1. Introduction

Transporters are essential transmembrane proteins that mediate the selective translocation of nutrients, metabolites, ions or drugs across cell membranes. Their function is related to cell nutrition, communication, homeostasis and response to drugs and stress. Consequently, their malfunction is associated with genetic or metabolic diseases and drug resistance [1,2]. Given their importance in sensing the environment and cell communication, transporter regulation of biogenesis and turnover is highly regulated at multiple levels. In prokaryotes, regulation concerns mostly transcriptional induction versus repression in response to physiological or stress signals. In eukaryotic cells, in addition to transcriptional regulation, transporter cellular expression is controlled at the level of subcellular trafficking, endocytosis, vacuolar turnover or recycling to the plasma membrane (PM) via the endomembrane system and the trans-Golgi network (TGN) [3,4]. Furthermore, eukaryotes seem to employ several points of protein quality control safeguarding that transporters are properly folded during the first steps of their initial biogenesis or remain structurally ‘undamaged’ under specific stress conditions. In case transporters are misfolded or damaged by chemical or physical stress or as a result of prolonged activity, cells employ processes such as endoplasmic reticulum-associated degradation (ERAD) and proteasome degradation, as well as specific chaperone-assisted selective autophagy [5] in addition to endocytosis, for driving transporter turnover [6,7,8,9]. A critical aspect of transporter function, associated to their transmembrane nature, is their continuous and dynamic interactions with membrane lipids, a challenge not faced by soluble proteins [10]. Given that eukaryotes contain a plethora of different membrane lipids and that the overall profile of these lipids might change in response to nutrition and physiological or stress alterations, transporter biogenesis, function and turnover might also need to be regulated accordingly. Finally, as most eukaryotic cells are characterized by polarity (e.g., basolateral versus apical membranes), often associated to the presence of distinct membrane microdomains, transporters also face the challenge of targeting to the proper membrane or microdomain. The aforementioned brief account on the cellular, physiological or stress factors that affect transporter biogenesis, subcellular sorting and turnover, intends to point out the high number and apparent complexity of interactions of transporters with proteins involved in membrane trafficking and regulated degradation. Considering that the genomes of eukaryotic cells are predicted to encode nearly 7–10% of transporters (http://www.membranetransport.org/transportDB2/index.html accessed on 20 May 2021), a rough estimate of the number of proteins involved in solute, ion and drug cellular transport might approach 20–25% of a genome’s translating capacity.

One aspect of transporter cell biology that has been scarcely studied is the mechanism by which transporters are sorted to the PM of eukaryotes. In most cases, transporters are considered typical membrane cargoes that follow the conventional pathway of vesicular secretion [11]. In other words, transporters are thought to be co-translationally translocated in the ER [12] and sorted into specialized nascent microdomains called ER-exit sites (ERes), where they interact with components of the COPII complex (e.g., Sec23/Sec24-Sec13/Sec31). Subsequently, COPII vesicles carrying transporters, chaperones and v-SNAREs in their membranes bud-off, uncoat and fuse with early Golgi compartments (e.g., ERGIC or cis-Golgi), and ‘reach’ the trans-Golgi network (TGN) via Golgi maturation. Similar to most membrane cargoes, transporters are thought to be sorted out from the TGN in AP-1/clathrin-coated vesicles, via recruitment of the small GTPase Rab11. Subsequently, post-Golgi vesicles carrying membrane cargoes travel on microtubules to reach the vicinity of PM. The final step of cargo vesicle fusion in the PM is assisted by components of the so-called exocyst complex, transfer to actin filaments and the tethering action of t-SNAREs. However, our recent work had challenged this generally accepted, but formally unproven, view concerning sorting of *de novo* made transporters to the PM [8,13]. Using the ascomycete *Aspergillus nidulans*, an emerging model system for studying membrane trafficking, we provided evidence that several nutrient transporters traffic non-polarly to the PM via Golgi- and microtubule-bypass. Importantly, our finding suggests the existence of distinct subpopulations of COPII vesicles; one carrying polar cargoes which are secreted by the conventional Golgi/microtubule-dependent route, and one that drives transporter and possibly other non-polar cargoes directly to the PM. However, what distinguishes these two types of COPII vesicles, other than the cargoes included in them, remains elusive.

Another novel, and rather unexpected, finding concerning transporter cellular expression has also originated from studies of *A. nidulans* nutrient transporters. In brief, we have found that purine transporters distributed non-polarly along the entire length of the PM and polar cargoes involved in apical growth (e.g., chitin synthase, lipid flippases, etc.) use distinct mechanisms to be endocytosed [8]. In response to modification of the nutritional profile of the growth medium or due to prolonged activity, transporters are recognized by α-arrestin adaptors, which recruit a HECT-type ubiquitin ligase. Ubiquitination of transporters then orchestrates internalization in clathrin-coated vesicles, which are sorted via early endosomes to vacuoles for degradation. Surprisingly, in this process, clathrin-mediated endocytosis of transporters is independent of the function of AP adaptors (e.g., AP-2), which have been considered essential partners of clathrin function [14,15]. Contrastingly, polar membrane cargoes undergo clathrin-independent, but AP-2-dependent, endocytosis, followed by local recycling to the apical membrane [15]. Which proteins replace the role of clathrin as a vesicular coat in AP-2-dependent endocytosis of polar membrane cargoes, how clathrin drives transporter internalization in the absence of AP-2, or which molecular markers distinguish recycling early endosomes from those destined for vacuolar degradation, are issues that remain largely unknown.

To dissect the molecular details of the mechanisms driving the subcellular sorting of *A. nidulans* transporters, we considered applying unbiased proteomic approaches that might identify transient-specific interactions of transporters with their trafficking or turnover partners. An emerging powerful approach is that of ‘proximity dependent biotinylation’ (PDB) coupled with liquid chromatography-mass spectrometry (LC-MS/MS) [16,17]. Prior of using proximity biotinylation assays, we needed to develop and validate a proteomics LC-MS/MS-based protocol that would provide a detailed insight into the proteome of *A. nidulans* under specific conditions related to our goals. In particular, we wished to establish a protocol under which we would obtain the maximum representation of *A. nidulans* proteins that drive and control the biogenesis and turnover of transporters of our interest. The established proteomics protocol described herein led to identification of 5690 proteins, which to our knowledge is the highest number of proteins identified not only in *A. nidulans* but also in other Aspergilli and related fungi. This number represents ≈53% of the 10,687 theoretical open reading frames (ORFs) identified in AspGD or FungiDB, significantly increasing the number of previously characterized proteins. In their great majority, ‘missing’ proteins concerned non-essential primary and secondary metabolic processes and related regulators not expressed under the conditions we used. Most importantly, validation of the profile obtained versus proteins known to be essential for trafficking and turnover of membrane cargoes in *A. nidulans* or other model fungi strongly supports that the present proteomic analysis identified, in principle, all proteins necessary for membrane cargo trafficking and turnover.

## 2. Materials and Methods

### 2.1. Media, Strains, Growth Conditions and A. nidulans Transformation

Standard complete and minimal media for *A. nidulans* were used (FGSC, http://www.fgsc.net accessed on 5 May 2021). Media and chemical reagents were obtained from Sigma-Aldrich (Life Science Chemilab SA, Athens, Greece) and AppliChem (Bioline Scientific SA, Athens, Greece). A *ΔazgA ΔuapA ΔuapC::AfpyrG pabaA1argB2* mutant strain (ΔACZ) [18] was the recipient strain for transformation with the plasmid pAN510-UapA-TurboID. *azgA, uapA* and *uapC* encode major purine transporters [19], whereas *pabaA1* and *argB2* are standard auxotrophic mutations for para-aminobenzoic acid and arginine, respectively. Transformation was performed by generating protoplasts from germinating conidiospores as described previously [20].

### 2.2. Protein Extraction

Cultures of *A. nidulans* were grown in MM supplemented with 1% glucose and 10 mM NaNO_3_ as the nitrogen source at 30 °C for 21 h at 140 rpm. The mycelia were snap frozen in liquid nitrogen and pulverized in a mortar with a pestle. Fine powder (~250 mg) was dissolved in lysis buffer (4% *w*/*w* SDS, 0.1 M Tris-HCL, 0.1 M dithioerythritol (DTE) pH = 7.6), and lysis took place in a water bath sonication apparatus. After complete solubilization, centrifugation took place for 10 min, at 12,000 rpm and the remaining debris were discarded. The supernatant was precipitated with a combination of ice-cold methanol and ice-cold chloroform (1:4 *v*/*v*) for 24 h at −20 °C to remove SDS and concentrate proteins. The following day, the precipitant was centrifuged for 30 min at 3000 rpm, and the protein pellet was dried under vacuum. Extracted proteins were resuspended in lysis buffer and concentration was determined using the Bradford assay.

### 2.3. Enzymatic Digestion

*A. nidulans* whole proteins were reduced in 0.1 M DTE for 30 min and subsequently alkylated by addition of 0.05 M iodoacetamide in 8 M urea, 0.1 M Tris-HCL pH = 8.5 for 25 min at room temperature in the dark. Afterwards, a tryptic solution (500 μM) was added at a trypsin to protein ratio 1:100, and digestion took place overnight. The following day, peptides were subjected to spin filtration (Amicon 30 kDa units, Millipore, Bilerica, MA, USA), and the peptide solution was dried in a vacuum until complete dryness. The fine powder was re-dissolved in 100 μL of HPLC (Thermo Fisher Scientific, Foster City, CA, USA) phase A (99.9% water with 0.1% formic acid (*v*/*v*)). The final peptide mixture was cleaned from impurities by filtering through a Millex^®^ syringe-driven filter (Merk KGaA, Athens, Greece) unit and was then subjected to nano LC-MS/MS analysis.

### 2.4. One-Dimensional Liquid Chromatography (1D-LC) Separation

Peptide separation was performed using a reversed-phase analytical C-18 column (75 μm × 50 cm; 100 Å, 2-μm-bead-packed Acclaim PepMap RSLC, Thermo Scientific) and using an Ultimate-3000 system (Dionex, Thermo Scientific, Bremen, Germany) coupled to an LTQ-Velos Orbitrap Elite mass spectrometer (Thermo Scientific, Waltham, MA, USA). Firstly, 6 μL of peptide mixture were loaded on a C-18 pre-column at a constant flow rate of 5 μL/min in phase A. Total elution time for all runs was 360 min for a gradient of 2–35% phase B (99.9% acetonitrile, 0.1% formic acid) at a constant flow rate of 300 nL/min.

### 2.5. MS/MS Analysis

Mass spectra were collected in an Orbitrap Elite mass spectrometer fitted with a nano-spray source. The instrument was operated in a data-dependent acquisition mode with the XCalibur^TM^ v.2.2 SP1.48 (Themo Scientific, Waltham, MA, USA) software. Full-scan data were acquired on the 300–2000 *m*/*z* range with resolution set to 60,000 with a maximum injection time of 100 ms. Data-dependent tandem mass spectrometry for the 20 most intense ions per survey scan was performed with higher-energy collision dissociation (HCD) fragmentation in the Orbitrap at a resolving power of 15,000 and a collision energy of 36 NSE%. The resulting ion fragments were analyzed on the Orbitrap; MS/MS spectra were acquired with 15,000 resolving power and a maximum injection time of 120 ms. Measurements were performed using *m*/*z* 445.120025 as lock mass. Dynamic exclusion was employed within 45 s to prevent repetitive selection of the same peptide.

### 2.6. Data Analysis

The resulting MS/MS data were processed using Proteome Discoverer (version 1.4.0.388, Thermo Scientific, Waltham, MA, USA). MS2 spectra were searched with SEQUEST engine against in a database comprising 10,558 of *A. nidulans* proteins (UP000000560 UniProtKB, 10,558 protein entries, https://www.uniprot.org/ accessed on 29 January 2021). The search parameters applied were the following: two maximum missed cleavage points for trypsin; oxidation of methionine as a variable modification; 10 ppm peptide mass tolerance; and 0.05 ppm fragment ion tolerance. The validation of the peptide spectral matches was performed with the use of percolator based on q-values at a 0.01% false discovery rate (FDR). Additional peptide filtering was succeed based on Xcorr versus peptide charge values (percolator maximum Delta Cn was set at 0.05). Values of 2.2 for doubly-charged and 3.5 for triply-charged peptides were used. The minimum length of acceptable identified peptides was set as 6 amino acids.

### 2.7. Manual Curation and Protein Classification

The Proteome Discoverer software was used to retrieve annotation information for identified proteins from Uniprot Database. Subsequently, for each Uniprot accession, we searched the best matched gene accession number at the AspGD (http://www.aspgd.org/ accessed on 10 May 2021) using the BLAST search tool of the above database. Through an automated process proteins were assigned their Gene ontology (GO) terms (http://geneontology.org/ accessed on 20 May 2021), specifically GOs for topology and biological process were necessary for the purpose of our study. 

## 3. Results and Discussion

### 3.1. Development of a Protocol for Protein Extraction for Proteomic Analysis in A. nidulans

This work primarily aimed to survey the proteome profile of proteins expressed in young actively growing hyphae under conditions that favour the transcriptional expression of transporters of solutes than can serve as secondary N sources, such as nucleobases, ureides, amino acids, nitrate, nitrite and generally any nutrient that can be metabolized to ammonium [19,21]. A prerequisite for this was to grow *A. nidulans* in minimal media that do not contain a primary N source such as ammonium (or glutamine), known to repress the transcription of secondary N source transporters. In most cases, provision of a secondary N source also leads to transcriptional induction of the relative specific transporters (i.e., the presence of nitrate leads to induction to nitrate transporters, besides allowing significant expression via derepression of all other N source transporters). Given that catabolism of different secondary N sources leads to production of ammonium or glutamine, and thus to some degree of repression of secondary N source transporters, here we used one of the least repressing secondary nitrogen source, namely nitrate. To establish an optimized proteomics protocol under our conditions, we used a strain carrying mutations *ΔuapA*, *ΔuapC*, *ΔazgA*, *pabaA1* and *argB2*, transformed with vector pAN510-UapA-TurboID. This strain, which expresses a functional fusion of UapA with TurboID biotin ligase from a genome-integrated vector pAN510-UapA-TurboID, is designed for future experiments to identify interactions of the UapA transporter. 

Experimental planning of the protein extraction procedure was crucial in unearthing the *A. nidulans* proteome. Optimisation was performed in our general extraction protocol [22] in order to overcome the existence of the hard cell wall, high endogenous protease activity, and the tendency of hyphae to clump. The protocols tested consisted each of agents aiming both to disrupt the heavily fortified chitin walls and preserve protein integrity for further proper enzymatic digestion. Our main challenge was to ensure that proteins remain in-solution for subsequent digestion. To achieve this, we primarily used buffers containing chaotropes (such as urea and guanidine hydrochloride) that facilitate protein denaturation and unfolding in-solution. In order to avoid carbamylation of amino and sulfhydryl groups and further modify lysine and arginine residues caused by high urea concentration we added Tris-HCL (a cyanate scavenger) to the final solution. Use of protein extraction buffers based on chaotropic agents led to identification rates of 82–85% of the final *A. nidulans* protein list, reported herein. Further, an alternate protein extraction protocol was used that consisted of sodium dodecyl sulphate (SDS), a detergent that disrupts the non-covalent bonds in proteins, which improved protein extraction efficiency and solubilisation. Due to SDS not being compatible with MS since it suppresses ionization of peptides, a step was included where SDS was thoroughly removed from samples via precipitation with both chlorophorm and methanol (4:1 *v*/*v*). All the above trials led to adopting a final protocol using a high concentration (4% *v*/*v*) of SDS protein extraction buffer, followed by sequential water bath sonication steps in order to disrupt chitin walls and finally a two-solution precipitation procedure in order to entirely remove SDS from the solution as well as concentrate the proteins of interest. Our herein reported method is easily reproducible whilst effectively providing a accurate depiction of the proteomic profile of *A. nidulans*. Through our approach we deliver, by far, the most comprehensive overview of the specific fungus proteome. 

### 3.2. Proteome MS Analysis of A. nidulans Grown in Glucose MM Containing Nitrate as N Source

LC-MS/MS analysis identified in total 5690 *A. nidulans* proteins. We classified these proteins based on their function and subcellular localization, using the GOs annotated for each protein in AspGD (http://www.aspgd.org/ accessed on 20 May 2021) and Fungi DB (https://fungidb.org/fungidb/app accessed on 20 May 2021). Due to the fact that the vast majority of *A. nidulans* ORFs remains uncharacterized, information was withdrawn from orthologous proteins, mostly in *Sacharomyces* sp. In cases where one protein was involved in more than one biological processes, it was classified to the most suitable one, according to our judgement. The complete list of the 5690 proteins can be found on Appendix A. Notice that the actual number of unique proteins drops to 5413. The rest appear two or more times in the list, because several different Uniprot accession numbers identified herein correspond to the same protein. Figure 1 depicts as pie charts the putative subcellular localization and functional distribution of *A. nidulans* proteome. As illustrated in Figure 1a, there is a total lack of information about the topology of 33% of the identified proteins. As for the rest, the majority of them is located to the nucleus or/and the cytosol. Quite a high number of proteins is localized in mitochondria, while the percentage of proteins belonging to other organelles is less than 20% alltogether. Additionally, 1% of the detected proteins are found in two or more subcellular compartments (mixed topology). Concerning the categorization of proteins based on the biological process they are involved in, it is clear from Figure 1b that almost one fourth of *A. nidulans* proteome is of unknown function. A high percentage of proteins is related to house-keeping processes such as transcription, translation or RNA processing. Meanwhile, proteins involved in hyphal growth, sporulation, reproduction, cell diferentiation and cell cycle are less than 10%. In regard to metabolism, numerous proteins play a role in primary metabolic processes, while only 3% of the identified proteins are involved in secondary metabolite biosynthesis. This is rather expected, as primary metabolism is a fairly general category, including processes such as lipid and carbohydrate biosynthesis, protein degradation, DNA repair, cellular respiration, etc. Finally, proteins involved in transport of solutes across a lipid bilayer and macromolecules to a specific location reach 6% and 2%, respectively.

### 3.3. Transporters Identified

The list of transporters expressed under the conditions we used, as well as their putative substrate can be found on Appendix A. We identified 315 proteins annotated as putative primary (ATP-binding) or secondary (uniporters, cation symporters or antiporters) active transporters and ion channels in TransportDB (http://www.membranetransport.org/transportDB2/index.html accessed on 20 May 2021). This constitutes 44.5% of a total of 710 proteins tentatively involved in solute, metabolite, ion, or drug transport. Among the 710 transporters annotated, 611 (86%) are grouped in large evolutionary and structurally related families or superfamilies (https://www.tcdb.org/ accessed on 20 May 2021) made of >10 protein members. These are the Major Facilitator Superfamily (MFS) with 356 members, the Amino Acid-Polyamine-Organocation (APC) family and its related Amino Acid Auxin Permease (AAAP) group with 69 members, the mitochondrial carriers (MC) with 40 members, the nucleobase-related carriers NCS1/NCS2/CntA with 15 members, the ATP-binding Cassette (ABC) and the ATPase (P-ATPase and F-ATPase) superfamilies with 99 proteins, and finally 25 channel forming proteins. Apart from the 315 proteins annotated as putative or known transporters, we found 34 additional putative transporters (included in Appendix A), which have not been annotated before, to belong to the aforementioned superfamilies. As mentioned previously for the whole proteome, the actual number of unique transporters decreases to 338, taking into consideration that 11 proteins are represented twice in this list. Figure 2 depicts, as a pie chart, the main families of transporters found in Appendix A.

Below we highlight the analysis of major transporter families and selected proteins identified in our proteome analysis, giving particular emphasis to transporters of our interest. 

#### 3.3.1. Sugar and Related Drug Efflux Transporters

Sugar transporters fall within the MFS, which also includes proteins mediating the uptake (uniport of symport), exchange (antiport) or efflux of a plethora of other solutes, such as vitamins, polyols, drugs, Krebs cycle metabolites, phosphorylated glycolytic intermediates, peptides, osmolytes, siderophores, nucleosides, organic and inorganic anions, etc. Among the 145 unique MFS identified herein we found the functionally characterized high and low affinity glucose transporters MstA/HxtD/AN8737 [23,24], MstE/AN5860 [25], as well as, other characterized sugar-related transporters, namely MstD/HxtC/AN10891 (glucose and xylose; [24]), LacpA/AN3199 (galactose and lactose; [26]), XtrD/AN0250 and XtrE/AN4148 (xylose; [27]). We also identified several more transporters putatively selective for a variety of sugars, such as fructose (AN2794), rhamnose (AN8464), α-glucosides (AN7866), glycerol (AN8467), maltose (AN7067, AN3515), but also quinate (AN5734). We did not detect other previously partially studied glucose-related transporters, such as HxtA/AN6923, MstC/HxtB/AN6669, HxtE/AN1797, MstB/AN2475 [24,28] or transporters related to cellobiose (AN2814) or fucose (AN5742) [29]. Notice that the expression of several *A. nidulans* glucose transporters (e.g., HxtB, HxtC and HxtE) in yeast restores growth on glucose, fructose, mannose, or galactose, indicating that these transporters accept multiple sugars as substrates. Overall, among the 145 MFS identified the great majority concerned members that might well be specific for glucose and other sugars. Notice; however, that amino acid sequence identities cannot be used to rigorously predict whether an MFS is a *bona fidae* sugar transporter. On the other hand, MFS members associated with sugar or other nutrient uptake can be tentatively distinguished, based on amino acid sequence similarity, from those involved in the efflux of drugs, mycotoxins, antibiotics or other toxic compounds, as well as siderophores. We thus identified at least 19 proteins that are annotated as efflux transporters. These include proteins that might be involved in the export of aflatoxin (AN5370/AN7200), fluconazole (AN5329/AN4119/AN8089), aspyridone polyketide (AN8413), antibiotics (AN8366) and drugs (AN4481, AN1031, AN11217). Still, other MFS, also tentatively associated to the efflux of similar toxic metabolites (AN0970, AN7295, AN0018, AN2643, AN3254, AN9219, AN9382, AN7898, AN2840), or NmeA/AN2746 associated to efflux of toxic amounts of nucleobases and Cd [30], were not detected in our analysis. Finally, we detected several known major siderophore efflux transporters, such as MirA, MirB, and MirC [31].

From the above results we conclude that *A. nidulans* grown in minimal media supplemented with glucose as sole C source, besides expressing as expected major glucose carriers, it also expresses several MFS that might contribute to glucose or other sugar uptake. Some of these ‘extra’ sugar transporters might serve as constitutive sensors of sugars in the growth medium. In addition to sugar importers, a small number of efflux-related MFS are also expressed. These are probably related to detoxification of toxic metabolites, excretion of fungal antibiotics, and iron scavenging, accumulated when glucose is the major C source.

#### 3.3.2. Amino Acid Transporters

Amino acid transporters fall within two major, distantly related, groups—APC and AAAP. In *A. nidulans* there are 56 APC and 14 AAAP proteins annotated according to TransportDB. The APC transporter superfamily includes members that function mostly as PM H^+^-coupled symporters or antiporters of amino acids and related solutes, such as GABA, neurotransmitters, choline, putrescine, etc. In our proteome analysis we detected 18 annotated APCs (32.1%), plus one more (AN10905), possibly acting as a GABA transporter, which has not been annotated before. In *A. nidulans* and other Aspergilli, only three APC have assigned functions and specificities, namely PrnB (proline; [32]), AgtA (glutamic and aspartic acid; [33]), and GabA [34]. Only PrnB was not expressed under the conditions of the present proteome analysis. This is rather expected given that transcription of the corresponding gene is inducible only in the presence of the relative amino acid in the growth medium. We also detected six AAAP (42.8%). No AAAP has been functionally characterized in *A. nidulans*. Fungal AAAPs in yeast (Avt1-7) are known to import or export of amino acids from vacuoles in response to the N source supplied [35]. The AAAP transporters detected herein probably serve a similar function in sequestration of biosynthesized amino acids in and out the vacuole. The expression of several PM APC in the absence of external amino acids is more difficult to rationalize. It will be interesting to investigate whether some of them act as high-affinity amino acid sensors, or exporters of specific biosynthesized amino acids, when these are accumulated at potentially toxic concentrations.

#### 3.3.3. Nucleobase/Nucleoside Transporters

All 15 nucleobase-related transporters in *A. nidulans* are extensively studied in our lab and constitute a priority in our scientific interest in transporters. They can be classified in three groups. The NCS1 family, made of subgroups FurA-G and FcyA-D, includes 11 functionally characterized H+ symporters specific for purines, pyrimidines and related drugs (e.g., 5-fluorouracil or 5-fluorocytosine) [36,37]. The NAT family (also known as NCS2) includes three extensively characterized major purine/H+ transporters (UapA, UapC and AzgA) [38,39]. The CntA transporter, which defines a single-member family, is a H^+^ symporter specific for all nucleosides [40]. There is also a protein that is similar to ENT nucleoside transporters (distantly related to MFS), but its function remains elusive in *A. nidulans* [40]. In our proteome analysis we only detected a single NCS1 nucleobase transporter, namely FcyC (AN7967), which; however, has no recognized transport function [36]. Among the homologues of FcyC that with previously assigned transport functions, which are: FcyB (cytosine-purines), FcyD (minor adenine transporter), FcyE (minor guanine transporter), FurA (allantoin), FurD (uracil), FurE (secondary transporter for uracil, allantoin, uric acid), FurF and FurG (minor uracil transporters), none was expressed under the growth conditions used. This is expected as NCS1 gene transcription is known to be undetectable in media that do not contain nucleobases. Consequently, FcyC detected herein is the sole exception, suggesting that it might recognize substrates other than nucleobases. Among the three known NAT/NCS2 transporters, UapA was identified because the strain used in proteomics contained multiple plasmid-borne copies of this gene fused with TurboID [41] integrated in its genome. The same strain also carried total genetic deletions of AzgA and UapC, thus their gene products were not detected. The reasons for using this strain are explained in the Introduction and in Section 2.

#### 3.3.4. Other Transporters

We detected all transporters involved in nitrate/nitrite uptake (NrtB/AN0399, CrnA/AN1008, NitA/AN8647) [42], given that we used nitrate as the sole N source in our analysis. We also detected the major acetate transporter AcpA (AN5226) [43] and several transporters tentatively specific for sulfate transport (AN3665, AN3157, AN4645, AN10387), glycerol (AN2096) or sucrose uptake (AN2289, AN2982). 

#### 3.3.5. Ion Channels

*A. nidulans,* as most fungi, has a small number of proteins, 25 in this case, which are known to be compatible with ion channel structure. These are annotated as selective for ammonium, various cations or water, whereas two are homologous to mechanosensitive ion channel of prokaryotes. In agreement with TranportDB, we detected thirteen proteins (52%), representative of most major channel subtypes, such as those selective for ammonium (MepB/AN0209, MepA/AN0209, MeaA/AN7463) [44], Cu^2+^ (AN3813), Mg^2+^/Al^2+^ (AN5856), the mitochondrial inner membrane Mg^2+^ channel (AN7826), vacuolar cation/calcium channels (Yvc1/AN3155, Cch1/AN1168, AN0610), the translocation protein Sec62 homolog (AN6269), the Annexin XIV homolog (AN2427), and both members of the Small Conductance Mechanosensitive Ion Channel (MscS) family (AN6053 and AN7571). We also found three more ion channels in our list that were not annotated before. These include putative potassium/calcium ion channels (Trp2/AN0566, AN7443, and AN10235). Interestingly, we did not detect aquaporins (AN0830, AN2822, AN3915, and AN7618), some of which seem to be expressed during germination (I. Kalampokis, G. Diallinas unpublished results).

#### 3.3.6. Mitochondrial Solute Transporters

*A. nidulans* has 40 annotated mitochondrial carriers (MC). Some of these proteins might be located in peroxisomes. MCs are involved mostly in antiport, but also H^+^-coupled symport, of keto acids, amino acids, nucleotides, inorganic ions and co-factors across the mitochondrial inner membrane. We identified 25 MC proteins (21 annotated in TranportDB), including carriers putatively selective for ATP/ADP, L-aspartate, L-glutamate, L-glycine, pyruvate, NAD, tricarboxylate, phosphate, thiamine, coenzyme A, pyridoxal 5′-phosphate (PLP), adenosine 5′-phosphosulfate (APS), 3′-phospho-adenosine 5′-phosphosulfate (PAPS), or Mg^2+^ export. Most of them have not been functionally analyzed in *A. nidulans.*


#### 3.3.7. Primary Active Transporters (ATP-Binding)

The ABC Superfamily (48 members) and the two groups of P-ATPase and F-ATPase (51 members in total) constitute the so-called ATP-binding primary active transporters in *A. nidulans*. ABC fungal transporters catalyze import and mostly efflux of metabolites, xenobiotics, drugs and lipids. Several ABC are thus related multidrug and antifungal resistance (MDR). The physiological role of ABC transporters involved in MDR is the export of lipids, lipopolysaccharides, lipoproteins, siderophores, toxins, peptides, or metals. Thirty-one out of the 48 annotated ABC transporters were detected (64.5%), all putatively related to efflux of toxic compounds. These include the functionally analyzed AtrA-D [45,46], which export azole fungicides, cycloheximide, chloramphenicol, several other antifungals and plant defense toxins, and may also be involved in the secretion of penicillin, as well as, AtnG that may excrete linear lipopeptide aspercryptin (secondary metabolite) [47]. We also found three more putative ABC transporters (AN3509, AN8813, and AN11934) that have not been annotated in TransportDB. This result shows that in minimal-glucose media *A. nidulans* expresses constitutively a significant number of efflux transporters for detoxification or excretion of secondary metabolites.

Fungal P-ATPases catalyze cation or H+ uptake and/or efflux across the PM or the ER (i.e., cation pumps), being involved in signal transduction, transcription regulation, cell growth, apoptosis, cell motility, pH or stress signaling, etc. They can also be involved in asymmetric phospholipid distribution in the PM (the so called flippases). We identified 11 out of 23 annotated P-ATPases putatively selective for sodium, potassium, or calcium, or acting as major phospholipid flippases. Among them is PmaA/AN4859, the major PM proton pump (H^+^-ATPase), which is an essential protein necessary for establishing a H+ gradient through which all solute transporters function [48]. In addition, we identified YgA/AN3624, a copper transporter necessary for development and conidia pigmentation [49], EnaB-C Na^+^ and Li^+^ pumps involved in tolerance to cation toxicity and adaptation to alkaline ambient pH [50], and several putative Ca^2+^ transporters (AN2827, AN5088, AN5743, and AN1189/PmcA). Interestingly, EnaB has been shown to accumulate at structures resembling the endoplasmic reticulum, rather than being a PM protein. 

F-type ATPases are found in fungal mitochondria where they use a proton gradient to drive ATP synthesis. In that sense, they are considered as ATP synthases, rather than transporters *sensu strictu*. F-ATPases are close relatives of the V-ATPases that couple the energy of ATP hydrolysis to proton transport across endosomes, lysosomes, and secretory vesicles, playing crucial roles for the function of these organelles and apparently protein and lipid subcellular trafficking in fungi. We identified 21 out of the 29 annotated F-ATPase transport proteins, of which twelve are predicted to be vacuolar and nine mitochondrial. Among these, VmaA, a target for bafilomycin or concanamycin, has been functionally characterized and shown to be critical for vegetative growth, mostly at pH > 7, conidiospore production, and hypersensitivity to Zn^2+^ [51].

### 3.4. The Profile of Proteins Involved in Cargo Membrane Trafficking and Turnover Validates the High Sensitivity of the Proteome Identified

To further validate the sensitivity of our protein extraction method and mostly the LC-MS/MS protocol followed, especially under the frame of our interest in transporter biogenesis and regulated turnover, we performed a ‘manual’ search of >100 proteins essential or highly critical for cargo membrane trafficking and turnover (Appendix A). In brief, we looked for the presence of proteins necessary for ERes/COPII biogenesis (Sar1, Sec23-Sec24, Sec13-Sec31), key proteins in Golgi functioning (SedV, GeaA, ArfA, Sec7, etc.), all Rab GTPases, SNARES, and tethers (Sec22, SedV, SynA, SsoA, etc.), all post-Golgi coats (e.g., H and L clathrin chains) and adaptors (e.g., all 4 subunits of AP-1 and AP-3, and GgaA), all components of the exocyst, ESCRT proteins involved in vacuolar cargo sorting, as well as all key endocytosis factors (e.g., cargo ubiquitination-related proteins, arrestins, PanA, Enta, EdeA, ApA, SlaB, SagA, Arps, AbpA, etc.). Impressively, all proteins essential for cargo biogenesis were indeed present in the proteome analyzed. The only exception is the product of AN11127, which was recently reported as the Sec12 protein of *A. nidulans* [52]. However, to our opinion, AN11127 might not be an orthologous protein to the Sec12 protein, which in yeast acts as an essential guanine nucleotide exchange factor (GEF) for activating Sar1p and initiation of COPII vesicle formation. Evidence against AN11127 being an isofunctional Sec12 orthologue comes from the observation that is has no amino acid similarity with Sec12 proteins, and importantly, is not present in several Aspergilli and most ascomycetes.

We detected 17 out of the 22 annotated ESCRT proteins and practically all essential proteins involved in endocytosis of PM cargoes (see Appendix A). In respect to endocytosis, we found that seven out of the 10 α-arrestins are not expressed in our conditions. This is rather expected as most of these proteins act as adaptors recruiting HulA^Rsp5^ ubiquitin ligase to specific nutrient transporters in order to initiate their internalization and vacuolar degradation, when preferred C or N sources (i.e., glucose or ammonium) are present in the growth medium [14]. Given the fact that in our protocol we used nitrate as the sole N source, rather than ammonium, in addition to glucose as C source, the absence of specific α-arrestins strongly supports that their activity is tightly regulated at a transcriptional level by the N or C source used. Transcriptional regulation of α-arrestins by glucose availability has been shown to control the expression of the high-affinity glucose transporter by endocytosis in yeast [53].

The three arrestins that were detected are PalF involved in pH signaling [54], ArtE involved in growth and differentiation [14], and ArtB of unknown function. The presence of ArtB in our proteome analysis predicts that it might be involved either in the fine regulation of nitrate transporters, but most probably, in the endocytic control of transporters mediating the uptake of secondary C sources, other than glucose.

We also looked for proteins associated to autophagy, as these are also related to transporter quality control and biogenesis [5]. We detected 33 autophagy-related proteins. Autophagy contributes to the maintenance of cellular homeostasis by facilitating recycling of amino acids of degraded proteins and by eliminating abnormal or damaged proteins, contributing to cellular quality control. Autophagy is known to be always active in different cells and at different physiological conditions, and further enhanced under N starvation or stress. Interestingly, while most key Atg proteins were detected, we did not detect Atg4 and Atg5, which are among important autophagic proteins. Note; however, that both Atg4 and Atg5 are non-essential genes in yeast. Null Atg4 and Atg5 mutants are unable to undergo nucleophagy and pexophagy, and display reduced mitophagy, micronucleophagy, and decreased viability during nitrogen starvation [55]. Given the high sensitivity of our proteome analysis in detecting essential proteins, the absence of Atg4 or Atg5 might indeed signify that they are little important in young, exponentially growing, cells of *A. nidulans*, at least under the growth conditions used herein.

Finally, we also detected many critical proteins involved in *A. nidulans* growth such as those involved in cytoskeleton functioning (e.g., ActA, TubA, kinesins, dyneins, etc.) and mitosis or cell cycle regulation (e.g., Nim and Bim proteins, septins), but as expected several proteins involved in asexual spore differentiation were not detected (e.g., BrlA, WetA, etc.).

## 4. Conclusions

Using a novel protocol for the isolation of *A. nidulans* proteins and downstream LC-MS/MS proteomics, we identified 5690 expressed proteins. To our knowledge this is the highest number of proteins identified by proteomics in *A. nidulans* and other Aspergilli to date. The proteins identified represent ≈53% of the 10,687 theoretical ORFs annotated in AspGD or FungiDB. Most importantly, validation of the proteome obtained showcased that tentatively all proteins known or predicted to be essential for membrane cargo trafficking and conventional secretion in fungi were identified. These include all components necessary for the formation and functioning of ERes and COPII vesicles (Sar1, Sec12, Sec23, Sec24, Sec13, Sec31, and Sec13), key proteins for vesicular passage of cargoes from the early and late Golgi (e.g., SedV, GeaA, ArfA, HypB^Sec7^ etc.), as well as, the entire set of essential for trafficking Rabs, AP adaptors, exocyst components, SNARE proteins and cytoskeletal factors. Three Rab-like proteins, annotated as RabX, RabT, and RabF, of unknown function, absent from several fungi, were not identified. This strongly suggests that their function is not essential for growth in glucose-minimal media but is probably related to specific growth conditions. In addition to proteins essential for sorting and trafficking of newly made membrane cargoes, we also identified all major players controlling endocytosis, vacuolar sorting or recycling (e.g., HulaA^Rsp5^, α-arrestins, H and L clathrin, AP-2, SagA^End3^, SalB^End4^, Epsin, AbpA, Rab5, Rab7, ESCRT components, etc.). We also identified major proteins controlling the turnover of proteins, such as those involved in UPR, ERAD-proteasome and autophagy. Overall, we did not detect the absence of any major protein essential for membrane trafficking. This confirms that our protocol identifies the entire set of proteins involved in membrane cargo biogenesis, trafficking and turnover, and is thus proper for identifying interactions with specific cargoes, such as transporters, using appropriate methodologies. We identified several transporters of all types. These belong mostly to MFS and APC groups, which represent the two largest superfamilies of ubiquitous transporters. Based on the annotated number of transporters, we did not detect ≈59% of MFS and ≈68% of APC members. This is to be expected, as the expression of a great number of MFS transporters is related to either specific physiological conditions related to primary metabolism, or secondary metabolism often linked silenced gene clusters under laboratory conditions, or the efflux of antibiotics and other xenobiotics and drugs. In particular, several MFS related to the uptake of C sources other than glucose are expected to be tightly repressed (e.g., transporters of disaccharides, fructose, carboxylic acids, etc.). Additionally, most APC transporters, commonly related to the uptake of amino acids and their derivatives are expected to be absent, since the transcription of the corresponding gene is only induced by the presence of the relative amino acid in the growth medium. The need for specific induction of transporters is exemplified by the identification of all three nitrate/nitrite carriers, since in our protocol we used nitrate as the sole N source of the growth medium.

In line with the belief that our method led to highly efficient identification of all proteins essential for growth, we also detected key players of house-keeping functions concerning transcription, post-transcriptional modifications, splicing, translation, and major post-translational modifications, including kinases and phosphatases, and all major enzymes involved in membrane lipid biosynthesis (ergosterol, sphingolipids, phospholipids), or proteins controlling membrane proton gradient and pH. Our protocol is technically little challenging, reproducible and seemingly highly efficient for detecting the full *A. nidulans* proteome, which is mandatory for identifying specific protein–protein interactions via appropriate methodologies.

## Figures and Tables

**Figure 1 jof-07-00560-f001:**
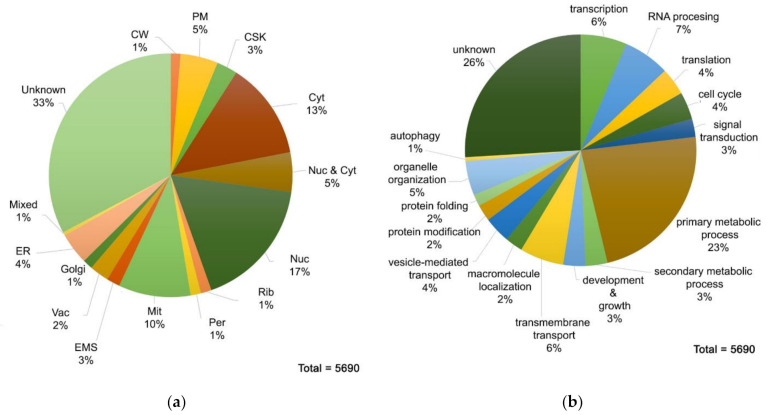
(**a**) Subcellular distribution of *A. nidulans* proteins. Abbreviations: Nuc, nucleus; Cyt, cytosol; PM, plasma membrane; CW, cell wall; CSK, cytoskeleton, Rib, ribosome; Per, peroxisome; Mit, mitochondrion; ER, endoplasmic reticulum; EMS, endomembrane system; Vac, vacuole. Notice the chart represents general categorization of proteins based in the majority of cases on the putative biological process as defined in AspGD and FungiDB. This means that several cytosolic proteins which might possess cryptic location signals might in fact be organellar. Thus, this pie chart is a gross representation of functional categories, as also appears in genome annotations. (**b**) Functional classification of *A. nidulans* proteins.

**Figure 2 jof-07-00560-f002:**
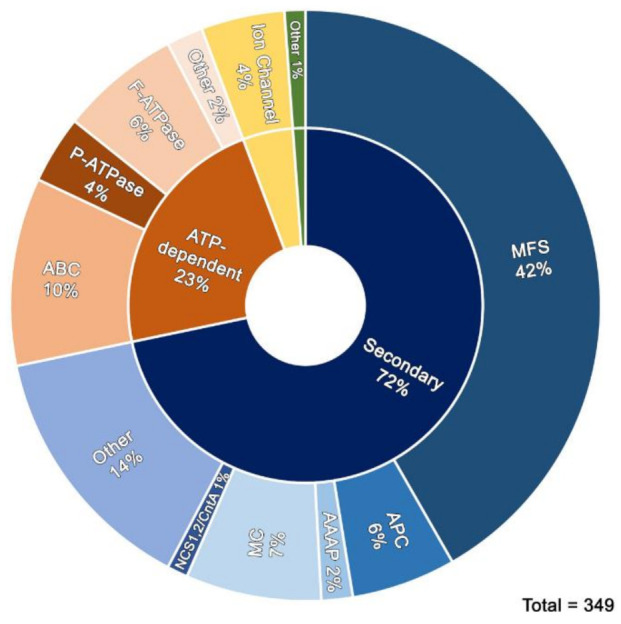
Pie chart of major transporter families.

## Data Availability

Data is contained within the article.

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
