# Peer review of "Profile of Membrane Cargo Trafficking Proteins and Transporters Expressed under N Source Derepressing Conditions in Aspergillus nidulans"

_jof, 2021, doi:10.3390/jof7070560_

Round 1

Reviewer 1 Report

In the manuscript “Profile of membrane cargo trafficking proteins and transporters expressed under N source derepressing conditions in Aspergillus nidulans“, the authors analyse proteomic profiles of the common, laboratory-used, filamentous fungus Aspergillus nidulans, exposed to the nitrogen derepressing conditions. The authors analyzed proteome of delta-ACZ mutant, impaired in the purine transport. Finally, they classified identified proteins according to the main metabolic categories, regarding cellular trafficking and membrane transport.

 In my opinion, presented methodological work is interesting and it is important to the other fungal researchers how to efficiently isolate and identify proteins from rather difficult, cell-wall possessing fungi. The high-throughput proteomic analysis done here is impressive. I see few points concerning this work, which makes it unclear, as listed below:

The use of delta-ACZ mutant, instead of the wild type strain, is poorly justified, and the results might  not reflect the situation in the wild type strain and other fungi.

All of the possible functions of the transporters is should be describe more cautiously, unless they are characterized enzymatically, functionally and published. If not, the authors may write putative transporters from the MFS family or tetraoxyanion-transporting SulP family etc. For example: row 395 where several transporters were listed, but none is involved in sulfate transport (the main and the only one in the reference Glasgow-origin is AN2730, albeit absent in the results. AN10387 is non functional pseudogene encoding impaired protein). The same with ‘sucrose’ transporters: until is not published and characterized enzymatically, it is better to write putative/hypothetical one.

Figure 1a leading to misunderstanding, since numerous transporters possess cryptic location signals (PMID: 18851831, PMID: 33086570), undetectable in silico by software

Row 401: the sentence is unclear. What does it mean “distinct from functionally analogous..”?

Row 427: yeasts have different type of mitochondria (PMID: 33906412), hence the set of proteins, especially transporters/channels would be different, and comparison to the A. nidulans ones is unjustified.

Author Response

Reviewer 1

The use of delta-ACZ mutant, instead of the wild type strain, is poorly justified, and the results might not reflect the situation in the wild type strain and other fungi.

We understand the concern of the reviewer, but the use of delta-ACZ mutant, instead of the wild type strain, is critical for subsequent downstream application of the protocol developed for specifically studying interactions of transporters, as the one planned using Biotinylation Proximity Assays. Using this genetic background, we can introduce, via transformation and targeted homologous recombination in the endogenous locus, any version of the three missing transporters (UapA, UapC or AzgA), tagged with the necessary epitotes, such the Turbo biotin ligase, for detecting transporter transient interactions. For example, we can introduce wild-type and mutant transporter versions which are trapped in the ER or mutants that do not undergo endocytosis. Our lab has already a plethora of such UapA versions, the sorting of which to the PM is affected. In addition, we need to stress that the absence of three purine transporters definitely does not affect the overall proteome obtained, especially in relation to the key trafficking proteins expressed, something which has also been validated experimentally in our work by detecting a complete profile of key trafficking proteins.

All of the possible functions of the transporters is should be describe more cautiously, unless they are characterized enzymatically, functionally and published. If not, the authors may write putative transporters from the MFS family or tetraoxyanion-transporting SulP family etc. For example: row 395 where several transporters were listed, but none is involved in sulfate transport (the main and the only one in the reference Glasgow-origin is AN2730, albeit absent in the results. AN10387 is non-functional pseudogene encoding impaired protein). The same with ‘sucrose’ transporters: until is not published and characterized enzymatically, it is better to write putative/hypothetical one.

We thank the reviewer for this comment. We did our best to modify the text in any such case detected. In fact, by publishing the list of transporters detected we wish to get a feedback from people who work on Aspergillus transporters and welcome any comment or correction concerning the possible function and specificity of annotated transporters.

Figure 1a leading to misunderstanding, since numerous transporters possess cryptic location signals (PMID: 18851831, PMID: 33086570), undetectable in silico by software

Figure 1a shows a general categorization of proteins based on the putative biological process as defined in AspGD and FungiDB. We agree with the reviewer that there are several cytosolic transporters possess with cryptic location signals. So this pie chart is a gross representation of functional categories, as also appears in genome annotations.  The full list of individual transporters can be found in Supplementary Table 1 and we welcome, as stated above, any comment concerning specific transporters. Given the aforementioned, we modified the text refereeing to Figure 1a to include reviewer’s point.

Row 401: the sentence is unclear. What does it mean “distinct from functionally analogous..”?

We wanted to stress that these transporters are not MFS, but to simply thigs we deleted the sentence “..all  structurally distinct from functionally analogous MFS transporters”.

Row 427: yeasts have different type of mitochondria (PMID: 33906412), hence the set of proteins, especially transporters/channels would be different, and comparison to the A. nidulans ones is unjustified.

The reviewer is right, we deleted the sentence “… but several yeast orthologues are critical for viability”

Reviewer 2 Report

Dear authors,

In this ms, the authors described a novel mass spectrometry based proteomics protocols adapted to conditions allowing for maximal identification of proteins related to N source uptake in A.nidulans. The aims of this study is to dissect the traffiking and mechanisms underlaying transporter subcellular biology. A data set of 5690 proteins have been identified which constituate the highest number of proteins identified representing around 53% of the total ORF.

But, the ms is more descriptive, no further analysis of potential transporter have been investigated. This ms is more appropriate to be published in a technical Journal but not enough to be published in JoF

Best regards

Author Response

Reviewer 2

But, the ms is more descriptive, no further analysis of potential transporter have been investigated. This ms is more appropriate to be published in a technical Journal but not enough to be published in JoF.

To our opinion the comment of this reviewer is not valid for the flowing reasons.

  1. Our ms was sent for reviewing which means that the all 3 inviting editors did not consider it out of the scope of JoF.
  2. Even if the reviewer considers our work “technical” this is not out of the scopes of JoF, which welcomes manuscripts on “Fungal Applied Technology”
  3. Our work indeed presents a novel technical approach for fungal proteomics, but as reviewer 1 also stresses, it describes a valuable method and provides important new findings (i.e. a most complete proteome of A. nidulans) to the other researchers working on Aspergilli and filamentous fungi in general.

Round 2

Reviewer 2 Report

The authors arguments presented and corrections are sufficient for the article to be published in this journal